# Lateral Cephalometric Analytical Uses for Temporomandibular Joint Disorders: The Importance of Cervical Posture and Hyoid Position

**DOI:** 10.3390/ijerph191711077

**Published:** 2022-09-04

**Authors:** Horia Opris, Mihaela Baciut, Simion Bran, Florin Onisor, Oana Almasan, Avram Manea, Tiberiu Tamas, Sebastian Stoia, Armencea Gabriel, Grigore Baciut, Bogdan Crisan, Mihaela Hedesiu, Liana Crisan, Ioan Barbur, Daiana Opris, Cristian Dinu

**Affiliations:** 1Department of Maxillofacial Surgery and Implantology, Iuliu Haţieganu University of Medicine and Pharmacy, 400012 Cluj-Napoca, Romania; 2Department of Prosthetic Dentistry, Faculty of Dentistry, Iuliu Haţieganu University of Medicine and Pharmacy, 32 Clinicilor Street, 400012 Cluj-Napoca, Romania; 3Department of Radiology, Faculty of Dentistry, Iuliu Haţieganu University of Medicine and Pharmacy, 32 Clinicilor Street, 400012 Cluj-Napoca, Romania

**Keywords:** temporomandibular joint disorders, hyoid bone, cervical posture

## Abstract

The temporomandibular joint disorder (TMD) is a syndrome that affects the masticatory muscles and temporomandibular joint (TMJ). Its pathophysiology is not yet fully known. Cephalometric analysis is used for routine evaluation regarding orthodontic treatment and other purposes. The aim of this study was to assess if using cephalometric analysis and TMJ conservative therapy to evaluate the hyoid bone position and the cervical posture reduced symptoms in adults with TMDs compared to no intervention. The authors conducted a systematic review of the literature (PubMed, Cochrane, Web of Science, Scopus, and Embase) for clinical studies of TMDs with conservative treatment and lateral cephalometric analysis of the hyoid and cervical posture. To assess the risk of bias for non-randomized clinical trials ROBINS-I tool was used. Out of 137 studies found, 6 remained to be included. Most of them found a link between TMD and lateral cephalometric analysis, but there was a high risk of bias. This review found a possible link between TMDs, the neck and cervical posture. There is a benefit reported regarding the use of the lateral cephalometry as a treatment, but more extensive prospective randomized clinical trials are necessary to be able to draw definitive conclusions.

## 1. Introduction

Temporomandibular disorder (TMD) is a complex syndrome with an uncertain pathophysiology [1] that affects the masticatory muscles, the temporomandibular joint (TMJ), and surrounding bone and soft tissue. Common clinical symptoms are restricted mandibular range of motion, mastication muscle soreness, TMJ pain, associated joint noise with function, generalized myofascial pain, and a functional limitation or deviation of the jaw opening [2]. These symptoms affect 6–12% of the population [3] and prevalence is known to vary with age and sex. Peak occurrence comes between 20 and 40 years and more frequently affects premenopausal women [2,4].

Generally, TMDs can be divided into articular/intracapsular, or internal derangements (ID), and non-articular/extracapsular disorders. More than 50% of non-articular disorders are related to myofascial pain although they can also be chronic pathologies such as muscular strain, fibromyalgia or myopathies. Myofascial pain is thought to occur from parafunctional habits (e.g., bruxism, or clenching) and manifests in the masticatory muscles from where it radiates to the ears, head and neck. Spasms and functional limitations are also encountered.

This condition can be treated by combining nonsteroidal anti-inflammatory drugs (NSAIDs), occlusal guards, physiotherapy, muscle relaxants, and injectable local anesthesia/steroid combinations into the masticatory muscle at insertion points [5].

According to the diagnostic criteria for temporomandibular disorder, articular disorders or internal derangements are divided into inflammatory and noninflammatory arthropathies. IDs refer to several abnormal positions of the disc in the condyle and the articular eminence, which are subtypes of TMD [6,7].

Regarding evaluation and diagnostic methods, cephalometric analysis is used to study parameters such as the anteroposterior or vertical relationship between the maxilla and mandible, the sagittal skeletal characteristics and facial asymmetry. These parameters have also been studied in connection with TMD IDs or TMJ disc displacements [8,9,10].

Other studies have investigated the association between temporomandibular disorders, malocclusion patterns, benign joint hypermobility syndrome and the initial condylar position. Subjects were analyzed using Rocabado Temporomandibular Pain Analysis, Helkimo, index parameters, the Carter–Wilkinson modified test and a mounting cast with condylar position, and a mandibular position indicator (MPI). Anterior crossbite and condylar displacements in the vertical plane are risk factors for developing TMJ symptoms [11]. The major difference among all the assessment methods is that Rocabado evaluates the hyoid position and the degree of extension of the cervical spine, which may need physical therapy prior to any occlusal appliances or treatment [12].

The aim of this study was to evaluate the effectiveness of lateral cephalometric analysis in conservative TMD treatment.

## 2. Materials and Methods

### 2.1. Protocol Development and Reporting Format

The review protocol was developed before the start of this review and was registered with the PROSPERO database under ID 356298. The review protocol was developed under the PRISMA guidelines [9]. The focused PICO (Population, Intervention, Comparison, Outcome) question was the following: In the adult population with temporomandibular joint disorders (P) does using cephalometric analysis and TMJ conservative therapy (I) evaluate the hyoid bone position, and does cervical posture reduce the symptoms associated with TMDs (O) compared to no intervention (C).

### 2.2. Eligibility Criteria

A prior, inclusion and exclusion criteria were defined. Only articles written in English were considered eligible. Regarding the study design, randomized control trials (RCTs) and non-randomized control trials (non-RCTs) studies were included.

Inclusion criteria were patients of either sex diagnosed with temporomandibular joint disorders by imaging which included cephalometric analysis. Only conservative, non-surgical treatment of temporomandibular joint disorders were included.

The exclusion criteria were studies not written in English; cross-sectional, animal, in vitro or in silico studies; reviews; meta-analyses; case reports; conference proceedings; book chapters; letters to the editor; technical notes; or unclear or insufficient information for data quantification.

### 2.3. Information Sources and Screening

An electronic search was conducted through five databases (PubMed, Web of Science, Scopus, EMBASE), to identify all in vivo studies published in English up to June 2022, using the following search phrases “((temporomandibular joint disease) OR (temporomandibular joint disorder) OR (TMD)) AND (hyoid OR (cervical posture)) AND ((cephalometry) OR (lateral cephalogram) OR (cephalometric analysis))”.

Screening was conducted in two stages. Two reviewers independently reviewed the titles and abstracts. The same two reviewers received the full texts of eligible publications and evaluated them separately. Then, all papers that met the inclusion criteria were evaluated in depth. Included were also reasons for exclusion. Any disagreements between reviewers were handled via conversation, and if a decision could not be reached, a third reviewer was contacted.

### 2.4. Data Collection

Two reviewers retrieved the characteristics of the studies: author, year, country, research design, study duration, primary aims, diagnosis, workup, cephalometric analysis, kind of therapy used, follow-up, complications, excluded patients, and eliminated results

### 2.5. Outcome Measures

Primary outcomes: clinical evaluation of temporomandibular disorder, signs, and symptoms. Secondary outcomes: radiological post-operative assessment and improved cephalometric analysis.

### 2.6. Quality and Risk of Bias Assessments

For randomized studies, the risk of bias (RoB) 2 tools were implemented if any were selected for inclusion [13]. A quality assessment was undertaken according to the Risk of Bias In Non-randomized Studies-of Interventions ROBINS-I [14], which has seven domains: confounding, osteoarthritis, rheumatoid arthritis, history of jaw injury or bruxism, connective tissue disease, previous orthodontic treatment). In addition, participants had to declare the types of interventions they’ve had and any deviations from the intended interventions; other risks of bias were, missing data and the measurements of results in the selection of the reported outcomes. The judgment of bias was evaluated as low (low risk for all fields), moderate (low/moderate for all fields), serious (serious risk in at least one field, but not critical in any field), critical (critical risk in at least one field), and no information (no clear evidence that the study is at risk and there is a lack of data in one or more key fields). The overall risk assessment was judged according to the ROBINS-I recommendations [14].

Two reviewers (H.O. and D.O.) separately evaluated the risk of bias for these studies, and if any disagreement occurred, a third reviewer (M.B.) intervened.

## 3. Results

### 3.1. Study Selection

The computerized search resulted in 137 items, which were reduced to 63 after duplicates were eliminated. A manual search turned up no other articles, and 38 were excluded based on a screening of the title and abstract. Complete texts of the remaining 25 articles were obtained, and these were evaluated qualitatively (Figure 1).

### 3.2. Study Characteristics

Table 1 shows a summary of the characteristics of the included studies, all of which were non-RCT (*n* = 6) [16,17,18,19,20,21]. Three studies were prospective [16,17,21] and three retrospective [18,19,20] in design. The number of patients per study varied from 15 to 187, with a total number of 397 included in this review.

Every article used a standardized assessment scale for the TMDs, and each of the included studies had a similar aim: to find a link between the temporomandibular joint disorder and the head posture/cervico-vertebral area using conservative treatment. For the majority of patients, this was an occlusal stabilization splint; one used a mandibular advancement appliance [21]. There was no homogeneity regarding the follow-up period, ranging from 1 h to one year (Table 1).

Most studies concluded that there was a link between TMDs, the specific treatment and the cervical posture. One study [16], though, found no connection between the masticatory system and the TMDs.

In the research conducted by Huggare et al. [14], the primary therapy consisted of counseling, occlusal correction, lower jaw muscle exercises, or a combination. Cervical dysfunction (CMD) was assessed for all subjects. Subjects were given treatment for craniofacial dysfunction (CMD), and their natural head position cephalogram was recorded by the same equipment and person. The dysfunction group had significantly more increased craniovertical and craniocervical angulations (FOR/OPT) compared to the controls. The curvature of the cervical spine (OPT/CVT) showed a significant straightening after stomatognathic treatment.

All patients except one, who had mild dysfunction, received some benefit from the treatment. The etiology of CMD is multifactorial and includes occlusal interferences, emotional disturbances, general musculoskeletal disorders, and an impaired state of health. There were no significant differences between the dysfunction group and the healthy controls in incisor inclination, overjet, and overbite. Compared to the healthy patients, the dysfunctional group exhibited an obtuse cranial base and a lower posterior-to-anterior face height ratio. The anatomy of 16 individuals with stomatognathic craniomandibular diseases and their age- and sex-matched controls demonstrated that the mandible must be pushed anteriorly to provide occlusal support for the craniofacial cartilage.

The results supported the idea that craniomandibular diseases, head position, and facial morphology are related, suggesting that treatment of TMD needs to place more emphasis on the craniovertebral area and less emphasis on dental occlusion. By forcing the hyoid bone to fall back, a bite opening may obstruct pharyngeal airflow. To restore the size of the airway, the head should be in a more extended position.

In the study by Moya et al. [17] eight women and seven men volunteered to participate. They possessed natural teeth, bilateral molar support, and muscle spasms in the sternocleidomastoid and upper trapezius.

Each participant wore a thermopolymerizing acrylic full-arch maxillary stabilizing occlusal splint (with flat occlusal surfaces and uniform, simultaneous and multiple contacts at relation-centric points). In the central incisor area, splints increased vertical occlusion by 4–5.5 mm. Two lateral craniocervical radiographs revealed an upright posture without a head holder and in a self-balancing position.

The baseline radiographs were taken with the mandible in the maximum intercuspal position. After one hour, a second radiograph was performed with the bottom teeth in modest contact with the splint’s occlusal surface.

The same examiner took two X-rays with and without an occlusal splint to reduce methodological error. Both measurements were averaged. Comparing angular or linear dimensions with and without the occlusal splint was done using statistical analysis. OPT/CVT did not see a statistically significant change. The splint raised distances D1 and D2, but not D4 and D5.

In the end, the occlusal splint affected craniocervical relationships significantly. Cervical hyper-extension and reduced cervical spine lordosis supported this. Changes in upper cervical spine lordosis indicated the need to evaluate periodically changes in craniocervical relationships after insertion of the occlusal splint.

Kang et al. [18] studied the association between the migraine and TMJ disfunction. The effects of persistent orofacial pain with the co-occurrence of TMD, migraines and cervical dysfunction were observed. Modification of the peripheral nociceptive input or the central sensitization process may be the focus of treatment efforts for TMD. Migraine and TMD symptoms may worsen because of cervical spine mobilization.

Current research is aiming to clarify migraine, cervical myofascial pain, and head and neck posture responses to orofacial pain therapy in TMD and migraine patients. At least five episodes meeting these criteria confirmed a migraine: a headache lasting 4–72 h or unilateral, pulsating, moderate-to-severe discomfort.

The questionnaire measured neck discomfort, and cephalometric analysis assessed craniofacial traits and head and neck position. TMD patients with orofacial discomfort and parafunctional behaviors were treated with splint treatment and physical therapy. Over eight hours of daily splint usage was deemed legitimate.

According to the principal results of the current research, the impact of orofacial pain management on the symptomatic improvement of painful TMD comorbidities, such as migraine and cervical dysfunction, seemed to be affected by the onset order of the symptoms of painful TMD.

Sensitization can result from a migraine, TMD, and neck pain, but interrupting peripheral stimulation can treat pain disorders and comorbidities. TMD patients had orofacial pain, neck discomfort, and a forward head position. Merging trigeminal and cervical nerve fibers may cause neck pain. This was the first study to examine the effect of TMD comorbidity on discomfort.

A recent study by Kang (2020) [19] investigated the associations among progressive temporomandibular joint osteoarthritis (TMJ OA), airway dimensions, and head and neck posture. TMJOApro seemed to develop forward head position more than TMJOAnopro or TMDnoOA. Patients with retrognathic facial profiles may exhibit decreased airway dimensions and altered head and neck posture. TMJ OA may cause severe TMJ condylar degeneration, resulting in a backward-positioned jaw and a hyperdivergent facial appearance. Patients with bony abnormalities on at least one side from TMJ CT were divided into two groups.

Erosive changes in the TMJ condyles or in combination with proliferation and deformed contours were regarded as indicators of the progression of TMJ OA. The presence of neck pain was assessed using the NDI, self-administered questionnaire. Cephalometric landmarks and variables were used for analysis of the head and neck posture. A supine position was more prominent in TMJOApro. The forward head posture (FHP) seemed to be more progressed in TMJOApro compared to the other two groups.

Patients with TMJ OA who have irregular occlusal contacts may have reduced pharyngeal airway capacity as well as altered cervicoposture. The purpose of this study was to determine the relationship between face structure, airway size, and posture in patients with TMJ OA. The jaw, which is positioned backward, and tongue may be important in reducing oropharyngeal airway capacity. In individuals with developing TMJ OA, the maxillo-mandibular connection changed dramatically.

FHP became remarkable in patients, which may have resulted from an attempt to compensate for decreased airway volume in the upright position.

In a 2019 study by Kim [20] the authors inspected the presence of abnormalities in the upper cervical vertebral spine (C1, C2, and C3) and craniofacial morphology in patients with TMDs. They were classified as positive or negative for joint disease based on the presence of abnormalities including osteophytes, sclerosis, and subchondral cysts.

Upper cervical spine anomalies were identified in patients who had fusions or posterior arch deficits (PADs), and lateral cephalograms were used to assess head and neck position. People were identified with TMD and given conservative treatment: counseling, stress management, NSAIDS, and rehabilitative exercise. After one year of therapy, final clinical evaluations and CBCT examinations were performed.

Data regarding the clinical characteristics of patients with and without upper cervical spine abnormalities (TMD) were analyzed. Those who had positive responses to neck muscle palpation had longer pain duration and more positive response in masticatory muscles (*p* = 0.002). The presence of TMD did not differ in the presence of any upper cervical abnormalities.

Other studies showed no influence of age, sex, or race on the occurrence of upper cervical vertebral anomalies. A patient with PAD tended to have a smaller mouth opening regardless of treatment although other indices reflecting TMD severity improved after treatment. The results of this study did not show significant differences in the presence or absence of upper cervical abnormalities in TMD patients, compared to those with undiagnosed TMD. However, cervical abnormalities may play a role in the development and maintenance of TMD symptoms.

This study was the first to evaluate differences in clinical symptoms and long-term treatment outcomes for TMD according to the presence of upper cervical abnormalities. In individuals with temporomandibular problems, upper cervical spine anomalies are associated with worse therapy. In a 2014 study by Santander et al. [21], the effects of a mandibular advancement appliance (MAA) on cervical lordosis in TMD patients with cervical pain was assessed. Cephalometric and clinical investigations were done both at the start and conclusion of the six-month trial period. A physiotherapist assessed posture impairment and implemented a postural re-education program. Twenty-two women with a clinical TMD diagnosis and chronic cervical pain were selected and underwent a three-month program of postural re-education. Exclusion criteria included patients with a history of arthritis, those undergoing spinal surgery or showing signs of early-onset dementia.

Each patient received an MAA constructed of clear, thermocured dental resin that moved the jaw forward to eliminate TMJ blockage and allow for a larger mouth opening. The patients were told to keep using the device and to keep their mandible in the same position. Because no one withdrew from the therapy, the statistics applied to all 22 patients.

The anterior repositioning splints successfully reduced or eliminated joint pain and clicking and associated secondary muscle symptoms. A possible reason for this effect is that the splints may alter adverse loading in the joint and correct the pathologic disc position. TMD patients exhibited significantly more segmental limitations and report more tenderness during palpation of the shoulder and neck muscles.

### 3.3. Quality and Risk of Bias Assessment

The result of the evaluation is provided in Appendix A. Two studies [16,17] were found to have a critical risk of confounding bias due to the lack of data. Other serious risks were the selection of the participants, no consecutive patients and no period of inclusion. One study had a serious risk of bias due to missing data on the follow-up period [16].

All the included studies were assessed as having a moderate risk of bias in the measurement of the outcome due to the lack of a blinded investigator.

The overall assessment of the risk of bias according to the ROBINS-I tool [14] can be seen in Figure 2 and extended information is in Appendix A.

The authors were deemed qualified to assess the quality of this systematic review objectively using the AMSTAR 2 checklist [22] which resulted in a moderate quality review.

## 4. Discussion

This systematic review assessed the clinical benefits of the lateral cephalometric analysis on the treatment of patients with temporomandibular joint disorders. In recent years studies have shown a prevalence of TMDs of around 25–35% [23,24,25] in the young population.

Cephalometric analysis is widely used for the pre-orthodontic assessment [26], mandibular growth pattern, cervical vertebrae assessment [27], upper airway obstruction and adenoid hypertrophy [28]. Recent developments have also linked it with evaluating the hyoid and tongue position [28]. Moreover, for cleft lip and palate patients, the velar ascent and morphology can be measured [29].

The main objective of this systematic review was to assess if using cephalometric analysis to evaluate the hyoid bone position and the cervical posture in TMD patients improved the outcome and the prognosis of the TMJ conservative treatment.

There is also a very important key aspect regarding the lateral cephalometric radiography and the natural head position. It is very hard to reproduce the same position, so to draw a conclusion a single-center study design should be considered. Several research articles [30,31] have shown that the reproducibility of the natural head position is low.

Due to the heterogeneity of the studies and lack of clear, uniform protocol, even with diagnosis criteria [24], a meta-analysis could not be undergone.

The first documented uses of the lateral cephalometric analysis to evaluate the hyoid position and the cervical spine in regards to temporomandibular disfunction was made by Rocabado in 1982 [32] when he described it in his study and in his following research [12,33] added a connection between dentistry and physiotherapy. His analysis was, and still is, used to facilitate the evaluation of a dentist, orthodontist or surgeon concerning the hyoid position and cervical misalignment and to how to address them prior to any invasive treatment concerning the occlusion, alignment of teeth, TMJ or orthognathic surgery. Although very popular in the Hispanic scientific world, his analysis is scarcely presented in research papers regarding the effectiveness of his assessment of TMD treatment. As seen in this review, little clinical research stands up to the rigors of critical evaluation—poor design, confounding bias, retrospective analysis and an unblinded examiner.

There is a real need for proper study design—a prospective clinical trial, with blinded observer, consecutive patients to be able to evaluate if the preoperative hyoid position and cervical spine misalignment diagnosis and treatment brought any value to the TMD treatment.

## 5. Conclusions

Given the extent and limitations of this study, the authors concluded that there are possible links between TMDs and the hyoid and cervical spine position, but due to the lack of studies, poor quality design and high risk of bias in the research, a definitive conclusion and recommendations could not be drawn. Although there are clear indications that this assessment brought some degree of benefit to patients, randomized prospective clinical trials are needed to re-enforce these statements.

## Figures and Tables

**Figure 1 ijerph-19-11077-f001:**
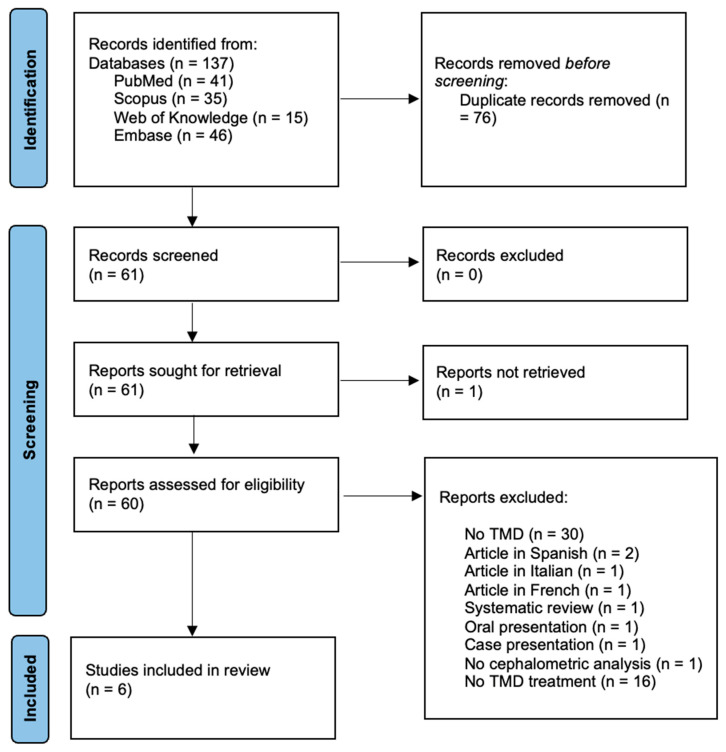
Prisma flow diagram of the review process [15].

**Figure 2 ijerph-19-11077-f002:**
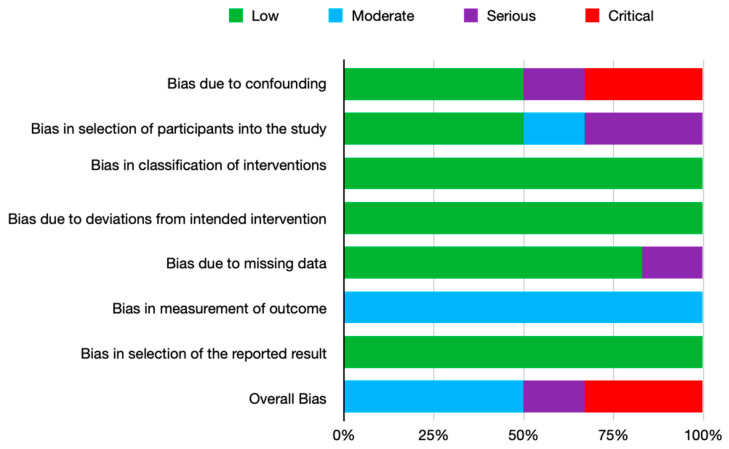
Summary of the risk of bias of included non-randomized trials with the ROBINS-I tool.

**Table 1 ijerph-19-11077-t001:** Summary of included studies with assessment of treatment.

Publication	Study Type	Age Inclusion Criteria: Male–Female Ratio, Age Range and Mean)	Type of Treatment	Outcome	Conclusions
[16]	P	*n* = 16 patients (M–F = 14:2 age range 21–40, mean age 26) Mean age 26 years 2 groups Control group investigated with no treatment *n* = 16	Counseling occlusal adjustment muscular exercises splint therapy	From 9 symptomatic patients, 3 remained symptomatic	The masticatory muscles, head muscles, and TMDs are closely related.
[17]	P	15 subjects (M–F = 8:7; Age range 20–41; Mean age 28.1) 2 groups: (1) natural dentition, muscle spasms in SCM and upper trapezius muscle (control); (2) same group after 1 h of splint therapy	Full-arch maxillary stabilization occlusal splint	Increase of the NSL-OPT angle Decrease of HOR/CVT, HOR/OPT, CVT/OPT Increase of distances D1 (C1–C6), D2 (C2–C6), D3 (C3–C6)	significant extension of the head on the cervical spine decrease in cervical spine lordosis
[21]	P	*n* = 22 female patients with TMDs Lordosis <20° muscle pain history for at least six months, and with an intensity >6 Patients had to present the angle formed by the posterior tangents to C2 and C7 of equal or less than 20°	six months of continuous MAA use	a significant increase in the cervical lordosis	homeostasis of the craniocervical system
[18]	R	*n* = 187 4 groups: (1) no TMD, *n* = 45; (2) painful TMD, *n* = 52; (3) painful TMD and then migraine, *n* = 47; (4) migraine and then painful TMD *n* = 43	Stabilization splint Physical therapy For 6 months	(4) improved less in orofacial, neck, and forward head posture after 6 months of TMD treatment than (2) and (3). After 6 months of TMD treatment, (4) had less migraine intensity, duration, and frequency than TMD1ST.	The onset order of comorbid conditions relative to TMD could determine the effects of TMD management on migraine and cervical dysfunction symptoms.
[19]	R	*n* = 114 M–F= 10:104 T0 T1 = 12 months Presence of osteoarthritis (1) TMDnoOA *n* = 28 (2) TMJOApro *n* = 45 (3) TMJOAnopro *n* = 41	stabilization splint physical therapy	In supine position, (2) had a larger oropharynx volume than (1), but there were no significant differences in the pharyngeal airway. T1 facial profiles (2) and (3) were more retrognathic than T0. (2) had a more forward head posture than (3) or (1).	TMJOApro may be related to upright head posture to compensate for reduced airway dimensions.
[20]	R	*n* = 43 TMD patients	Conservative therapy for 1 year	Before treatment, patients with cervical fusion (*p* = 0.019) or posterior arch deficiency (*p* = 0.004) had more neck muscle pain. After treatment, PAD patients had more mouth opening limitation (*p* = 0.028) and masticatory muscle pain (*p* = 0.014) than patients without the deficiency.	Upper cervical spine characteristics affect TMD treatment outcomes.

TMDs––temporomandibular joint disorders; HOR—true horizontal line; OPT—odontoid plane; TMJ—temporomandibular joint, OA—osteoarthritis; MAA—mandibular advancement appliance; P—prospective clinical trial; R—retrospective clinical trial; NSL/OPT—craniocervical angulation; HOR/CVT—true horizontal plane to cervical vertebrae tangent angle; HOR/OPT—odontoid process—true horizontal line angle; CVT/OPT—the cervical vertebrae—horizontal line angle; TMJOApro—progressive temporomandibular osteoarthritis; TMJOAnopro—no progressive temporomandibular osteoarthritis; TMDnoOA—without any pathologic bony changes in either side of the TMJ condyles.

## Data Availability

Not applicable.

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
