# Peer review of "Lateral Cephalometric Analytical Uses for Temporomandibular Joint Disorders: The Importance of Cervical Posture and Hyoid Position"

_ijerph, 2022, doi:10.3390/ijerph191711077_

Round 1

Reviewer 1 Report

Dear Authors,

first of all, congratulations on writing the presented work. The work is coherent and well written. The methods and results have been described quite comprehensively. However, the discussion and conclusions bring more questions than answers.
The paper is written correctly and quite interesting. However, the biggest weakness is that in order for this work to be of great value, the head positioning should be repeatable when taking a cephalometric radiograph. Unfortunately, this is not the reality. The conditions for taking such radiographs would have to be different, or they would have to be taken at a single center. While the head position is repeatable, the cervical position is often a forced position. Therefore, while it makes sense to take measurements of shape etc., it seems pointless to analyze distances and angles.

Best regards

Author Response

Dear Reviewer #1,

On behalf of all the coauthors I would like to express our consideration for your involvement and for the suggestions you have made to the article we have submitted. Following these recommendations, we have made the modifications accordingly, hoping to improve our paper to your expectations. Please find below the changes we have made to your requirement. 

  1. 1. Dear Authors,

    first of all, congratulations on writing the presented work. The work is coherent and well written. The methods and results have been described quite comprehensively. However, the discussion and conclusions bring more questions than answers.

The paper is written correctly and quite interesting. However, the biggest weakness is that in order for this work to be of great value, the head positioning should be repeatable when taking a cephalometric radiograph. Unfortunately, this is not the reality. The conditions for taking such radiographs would have to be different, or they would have to be taken at a single center. While the head position is repeatable, the cervical position is often a forced position. Therefore, while it makes sense to take measurements of shape etc., it seems pointless to analyze distances and angles.

Response: Thank you so much for your input and for your consideration. We also believe that the head positioning is very hard to be done properly even in a single center where there are multiple operators. We have added this point in the discussion paragraph (line 329).

Reviewer 2 Report

Dear Authors PICO question is not clear in the abstract and the material and methods section. Moreover many sentences, that are scientific or clinical affirmations  need citations. At my knowledge, Rocabado Analysis, is a practical method without any validation both in terms of sensibility and specificity related to TMD. But this is my ignorance please provide a reference. If a reference it is not provided I suggest to delete from the title the type of analysis otherwise a different approach to the SR has to be made. Recommended to report clearly the separate sections of PICO. In abstract, only the ROB tool for non-randomised studies mentioned. Which tool was used for randomised studies. Line 63-69, mentioned various analysis for TMD dysfunctions measurements. It is not clear why ROCABADO analysis required as compared to other analysis and what different information we get from rocabado analysis that we don’t get from other tool. The review should explicitly highlight and describe if the review was registered and the protocol was developed before actually writing this review. Please provide any justifications of any deviations of the review. This review lacks some significant information that needs reporting to the guidelines and checklist set in AMSTAR-2. Therefore, recommend to follow the checklist of AMSTAR-2 for quality reporting.

Author Response

Dear Reviewer #2,

On behalf of all the research team, I would like to thank you for your valuable feedback and for your appreciation. Finally, we are extremely thankful for your feedback. Please find below the changes we have made to your requirement. 

  1. 1. Dear Authors PICO question is not clear in the abstract and the material and methods section. Moreover many sentences, that are scientific or clinical affirmations need citations. 

Response: The authors have modified the abstract section to include all the aspects of the PICO question.

  1. 2. At my knowledge, Rocabado Analysis, is a practical method without any validation both in terms of sensibility and specificity related to TMD. But this is my ignorance please provide a reference. 

Response: Yes, you are very right. That is actually the basis of this research material. To our knowledge also Rocabado has a lot of clinical use but no actual validation from the standpoint of research soundness. Rocabado has shaped a lot of the clinical practice of the orthodontists, maxillofacial surgeons, and physical therapists who work with the TMDs. We believe that there is a place for his analysis in the research community and we are trying with this research article to objectively evaluate the current state of the research. There is a bridge and a gap which Rocabado tried and we believe , successfully, between physical therapy and malocclusion and this analysis objectively and easily makes the dental clinician evaluate and recommend physical therapy to modify the cervical spine, hyoid bone and occlusion.

Unfortunately, it does not provide enough quality studies and conclusions to make any assumptions and recommendations, although there are many actual clinical applications.

  1. 3. If a reference it is not provided I suggest to delete from the title the type of analysis otherwise a different approach to the SR has to be made. 

Response: We reconsidered the title to meet your recommendations.

  1. 4. Recommended to report clearly the separate sections of PICO. 

Response: we have modified the PICO question to separate the sections more clearly. 

  1. 5. In abstract, only the ROB tool for non-randomised studies mentioned. Which tool was used for randomised studies. 

Response: I would like to appreciate the attention to details that you put into this review. The authors have decided not to include. Unfortunately, there were no randomised studies to be included so we opted not to include what tools should have been used if any studies were to be included. We prefer RoB 2 for randomised studies.

There are no randomized studies to be included so we decided to only mention the tool which was used for the non-randomised studies.

  1. 6. Line 63-69, mentioned various analysis for TMD dysfunctions measurements. It is not clear why ROCABADO analysis required as compared to other analysis and what different information we get from rocabado analysis that we don’t get from other tool. 

Response: We have added a paragraph with citation to specify what the Rocabado analysis does different.

  1. 7. The review should explicitly highlight and describe if the review was registered, and the protocol was developed before actually writing this review. 

Response: The protocol was developed before writing this review and the Prospero number is 356298.

  1. 8. Please provide any justifications of any deviations of the review. This review lacks some significant information that needs reporting to the guidelines and checklist set in AMSTAR-2. Therefore, recommend to follow the checklist of AMSTAR-2 for quality reporting.

Response: Thank you so much for the suggestion. We have applied the checklist mentioned and have added a paragraph in the results to objectively assess the systematic review.

Reviewer 3 Report

In reviewing the manuscript, there are some areas of clarification:

In the introduction: Need to make clearer that treatment of the TMJ will be evaluated.

The study characteristics are confusing to read. I would recommend rephrasing a lot of paragraphs and making them shorter.

In your results: Pag Line 178-179 repeated sentence

The manuscript is clearer when short, concise statements are used rather than longer sentences or sentence fragments.

It is risky to say that there is a possible link since you didn’t find any strong relation, and there is a lot of bias in the literature.

Author Response

Dear Reviewer #3,

On behalf of all the research team, we would like to welcome and appreciate research soundness to which your observations helps us improve on the quality of the manuscript. Finally, we are extremely thankful for your feedback. Please find below the changes we have made to your requirement. 

  1. 1. In reviewing the manuscript, there are some areas of clarification: In the introduction: Need to make clearer that treatment of the TMJ will be evaluated.

Response: we have modified to specifically include the conservative treatment of the TMDs (exclude the surgical treatment.

  1. 2. The study characteristics are confusing to read. I would recommend rephrasing a lot of paragraphs and making them shorter.

Response: we have rephrased the results paragraph to be more easily readable and to be shorter.

  1. 3. In your results: Pag Line 178-179 repeated sentence

Response: We have corrected and deleted the sentence. Thank you!

  1. 4. The manuscript is clearer when short, concise statements are used rather than longer sentences or sentence fragments.

Response: We have rephrased the sentences to make them shorter and clearer.

  1. 5. It is risky to say that there is a possible link since you didn’t find any strong relation, and there is a lot of bias in the literature.

Response: Thank you so much for your comment. Yes, you are correct that there is a risk in saying that there is a possible link. At the same time, we cannot exclude a link. The basis of this study is that the Rocabado analysis is widely used in clinical practice by TMJ specialists such as orthodontists, maxillofacial surgeons and physical therapists with very good results. We cannot exclude that there is some benefit of using this method which was first brought up to light in the 1982 and still is in use. That’s why we assume the final statement that there is a possible link but it is hard to say using scientific righteousness how much and when there is a benefit of using this method.